# Nanoscale Heat Conduction in CNT-POLYMER Nanocomposites at Fast Thermal Perturbations

**DOI:** 10.3390/molecules24152794

**Published:** 2019-07-31

**Authors:** Alexander A. Minakov, Christoph Schick

**Affiliations:** 1Prokhorov General Physics Institute of the Russian Academy of Sciences, GPI RAS, Vavilov str. 38, 119991 Moscow, Russia; 2Institute of Physics and Competence Centre CALOR, University of Rostock, 18051 Rostock, Germany; 3Butlerov Institute of Chemistry, Kazan Federal University, 18 Kremlyovskaya Street, 420008 Kazan, Russia

**Keywords:** Nonequilibrium heat transfer, nanometer scale heat conduction, crystallization kinetics, ultra-fast calorimetry

## Abstract

Nanometer scale heat conduction in a polymer/carbon nanotube (CNT) composite under fast thermal perturbations is described by linear integrodifferential equations with dynamic heat capacity. The heat transfer problem for local fast thermal perturbations around CNT is considered. An analytical solution for the nonequilibrium thermal response of the polymer matrix around CNT under local pulse heating is obtained. The dynamics of the temperature distribution around CNT depends significantly on the CNT parameters and the thermal contact conductance of the polymer/CNT interface. The effect of dynamic heat capacity on the local overheating of the polymer matrix around CNT is considered. This local overheating can be enhanced by very fast (about 1 ns) components of the dynamic heat capacity of the polymer matrix. The results can be used to analyze the heat transfer process at the early stages of “shish-kebab” crystal structure formation in CNT/polymer composites.

## 1. Introduction

Recent progress in the synthesis of nanomaterials requires a deep theoretical and experimental study of the thermal transport on the nanometer scale. Advances in ultrafast nanocalorimetry stimulate experiments with ultrafast temperature changes at rates up to 10^7^ K/s. The experiments using ultrafast nanocalorimetry provide opportunities to study phase-transition kinetics at microsecond and shorter time scales in micro- and nanoscale objects [1,2,3,4,5,6,7,8]. Technologically important polymer nanocomposites have been investigated recently by ultrafast nanocalorimetry [6,7,8]. However, the classical heat conduction theory is insufficient for ultrafast processes in nanocomposites if the local temperature is varying suddenly [9,10,11,12]. In addition, polymer-based nanocomposites have an interesting specificity for fast thermal perturbations [13,14]. In fact, relaxation processes associated with the dynamic heat capacity $c_{dyn}\left(t\right)$ of polymer-based systems are considerable at fast thermal perturbations [15,16,17,18,19]. Indeed, the spectrum of relaxation times of thermal excitations in polymers is extremely wide, which is proved by experiments on broadband dielectric spectroscopy and heat capacity spectroscopy [15,16,17,18,19,20,21,22,23,24,25,26,27,28,29,30,31]. Molecular motions in polymers are very complex, especially in the amorphous polymer phase [32,33,34,35]. This leads to the effect of temporal dispersion of heat capacity in polymers and organic liquids [18,19,23,24,36,37,38,39,40,41].

The temporal dispersion of the heat capacity of a polymer matrix can strongly influence the heat transfer in polymer-based nanocomposites. Nanocomposites with carbon nanotubes (CNT) are very important for many applications. The aim of this article is to study the nonequilibrium thermal response of the polymer matrix to fast local thermal perturbations around CNT in polymer/CNT nanocomposites. Our goal is to solve the heat transfer problem of local thermal perturbations around CNT. These thermal perturbations can occur in the early stages of the formation of crystal structures in CNT/polymer composites. The crystal structure in CNT/polymer composites has a “shish-kebab” geometry [42,43,44]. Indeed, the local temperature in the region of crystal birth can be significantly increased due to the heat released at crystallization even under isothermal boundary conditions for the whole sample. In this paper, we focus on the analytical solution of the problem with *dynamic* heat capacity $c_{dyn}\left(t\right)$ at *nonequilibrium* thermal response of the polymer matrix.

In fact, the local temperature in the polymer matrix with dynamic heat capacity can be much more overheated than in the equilibrium case at early stages of the fast heating process [13,14]. Local overheating in the early stages can significantly affect the process of crystallite formation since the thermodynamic parameters, such as viscosity, considerably depend on temperature. It is interesting that even fast components of the dynamic heat capacity (with relaxation time $\tau_{0}$ about 1 ns) are significant [13,14]. In the present work, we focus on the dynamics of the temperature distribution T(t,r,z) around CNT at nanosecond and longer time scales. The thermal response of the polymer matrix around individual CNT under pulse heating in cylindrical geometry is considered. The effect of thermal-contact conductance of the polymer/CNT interface and CNT parameters is studied. Specific heat capacity at constant pressure $c_{p}$ is discussed below, but the index p is omitted further.

## 2. Heat Conduction in Polymer Matrix with Dynamic Heat Capacity

This paper focuses on thermal transport in nanocomposites with a dielectric polymer matrix at temperatures above the low-temperature range. Organic glass-forming polymers are often used as a matrix for nanocomposites. In the case of an amorphous polymer matrix, the matrix can usually be considered as homogeneous up to the nanometer scale. It is further assumed that the length scale of the thermal gradients $T{\left(\partial T/\partial x\right)}^{-1}$ is longer than the phonon mean-free-path in the polymer matrix. Thus, nonlocal effects [11] and the ballistic contribution to heat transfer in the polymer matrix can be neglected. The phonon mean-free-path in an amorphous polymer matrix is less than 1 nm [45,46,47,48,49], and the phonon excitations are relaxing on a time scale of 10 ps. In fact, the phonon distribution relaxes to equilibrium in the time interval Δt when the thermal-diffusion length $\sqrt{4D_{0}\Delta t}$ exceeds several phonon mean-free-paths [13]. Thus, Δt can be estimated at about 10 ps for an amorphous polymer matrix with $D_{0}$ of the order of 10^−7^ m^2^/s and a phonon mean-free-path about 1 nm. This relaxation time scale can be longer, up to 1 ns, in the case of crystalline polymers. In any case, the thermal conductivity can be considered as an equilibrium parameter at Δt> 1 ns [13,14]. In fact, the characteristic time constants describing the heat flux lag and the temperature gradient lag in the Maxwell–Cattaneo approach [9,10] associated with nonequilibrium behavior of the thermal conductivity are much less than 1 ns in amorphous polymers; for details, see Reference [13]. Therefore, the effect of non-Fourier heat conduction can be neglected on nanosecond and longer time scales. However, in glass-forming polymers, the effect of dynamic heat capacity provides a strong nonequilibrium contribution to the thermal response. In this paper, we focus on the nonequilibrium thermal response associated with the dynamic heat capacity of the polymer matrix. The effect of the dynamic heat capacity is significant for a wide range of relaxation times even on nanosecond and longer time scales when the thermal conductivity can be considered as equilibrium parameter. Thus, we consider nonequilibrium thermal response of the polymer matrix associated with the dynamic heat capacity. The Maxwell–Cattaneo approach associated with nonequilibrium behavior of the thermal conductivity can be significant at the picosecond scale and will be considered in a separate article. Thus, the diffusive heat conduction is considered further.

Next, the thermal parameters of the polymer matrix are considered independent from the temperature for small thermal perturbations. However, the temperature dependence of the relaxation time associated with the dynamic heat capacity is taken into account.

The temporal dispersion of the dynamic heat capacity of glass-forming polymers can be described similarly to the theory of dielectric permittivity dispersion [50,51]. Thus, heat transfer in the polymer matrix with the dynamic heat capacity $c_{dyn}\left(t\right)$ can be described by Equation (1)
(1)$$\frac{\partial }{\partial t}\int_{0}^{\infty }\rho c_{dyn}\left(\tau \right)\frac{\partial }{\partial t}T\left(t-\tau ,r\right)d\tau =\lambda \Delta T\left(t,r\right)+\Phi \left(t,r\right)$$
where Φ(t,r) is the volumetric external heat flux. In fact, Equation (1) follows from the diffusive parabolic heat equation if one takes into account the dynamic heat capacity of the glass-forming material [13,14]. Indeed, the local heat absorption at time *t* depends on the local temperature at previous times. Thus, the temporal dispersion of the dynamic heat capacity is described by the convolution integral (see Equation (1)), according to the linear response theory [50,51]. This equation can be used on at least nanosecond and longer timescales as well as on a length scale greater than 1 nm for an amorphous polymer matrix, as explained above. Equation (1) can be solved if the dynamic heat capacity $c_{dyn}\left(t\right)$ is known. Consider the base example. Assume that $c_{dyn}\left(t\right)$ obeys the Debye relaxation law:(2)$$c_{dyn}\left(t\right)=c_{0}\left(1-\varepsilon_{0}exp(-t/\tau_{0})\right)$$ where $\varepsilon_{0}=\left(c_{0}-c_{in}\right)/c_{0}$ and $c_{in}$ and $c_{0}$ are the initial and equilibrium heat capacities, respectively. In fact, $c_{dyn}\left(t\right)\to c_{in}\;\mathrm{at}\;t\to 0$ and $c_{dyn}\left(t\right)\to c_{0}\;\mathrm{at}\;t\to \infty$. Then from Equations (1) and (2), we get Equation (3) for cylindrical geometry and at zero initial condition: T(t,r,z)=0 if t≤0.
(3)$$\frac{\partial T}{\partial t}-D_{0}\left(\frac{\partial^{2}T}{\partial r^{2}}+\frac{1}{r}\frac{\partial T}{\partial r}+\frac{\partial^{2}T}{\partial z^{2}}\right)=\frac{\Phi \left(t,r,z\right)}{\rho c_{0}}+\varepsilon_{0}\frac{\partial }{\partial t}\int_{0}^{t}exp(-\tau /\tau_{0})\frac{\partial }{\partial t}T\left(t-\tau ,r,z\right)d\tau$$
where $D_{0}=\lambda /\rho c_{0}$. Note the upper limit of the integral in Equation (3) equals t since 0≤τ≤t at zero initial condition: T(t,r,z)=0 if t≤0. In fact, $c_{0}$ and $c_{in}$ are related to the heat capacities $c_{lq}$ and $c_{g}$ of the liquid and the glassy states of the polymer matrix, respectively. Thus, $\varepsilon_{0}$ is related to the ratio $\left(c_{lq}-c_{g}\right)/c_{lq}$. In polymers, this ratio can be in the range 0.2–0.3, as in polystyrene [18] and polyvinyl acetate [52]. However, this ratio can be considerably increased in ultra-stable glasses obtained by vapor deposition at temperatures below the glass transition temperature. Thus, in ethylbenzene, this ratio ranges from 0.35 to 0.52 depending on the deposition temperature [53]. As an example, the parameters $c_{0}=$ 2 × 10^6^ J/m^3^K, $c_{in}=\left(2/3\right)\cdot c_{0}$, and $\varepsilon_{0}=1/3$ are used for model calculations. However, the analytical solution presented in this paper can be applied to any glass-forming polymer matrix.

The dynamic heat capacity $c_{dyn}\left(t\right)$ is a monotonically relaxing function of time. Thus, $c_{dyn}\left(t\right)$ can be presented as a continuous sum of exponents [54,55]. Denote by $H\left(\tau_{0},T\right)$ the distribution function of the relaxation time $\tau_{0}$, then
(4)$$c_{dyn}\left(t\right)=c_{0}-\left(c_{0}-c_{in}\right)\int_{0}^{\infty }H\left(\tau_{0},T\right)exp(-t/\tau_{0})d\tau_{0}.$$

In fact, the distribution function $H\left(\tau_{0},T\right)$ can be found from the results of broadband heat capacity spectroscopy [18]. Therefore, T(t,r,z) can be represented as a linear combination of solutions of Equation (3) with different $\tau_{0}$, for details see [14]. Next, we consider the effect of one component of the dynamic heat capacity (with a certain $\tau_{0}$) on the dynamics of the temperature distribution in the polymer matrix around CNT. However, averaging over $H\left(\tau_{0},T\right)$ can be performed. The distribution function $H\left(\tau_{0},T\right)$ can be specified for a given polymer, as shown for polystyrene (PS) and poly(methyl methacrylate) (PMMA) [14].

## 3. Heat Transfer Problem for the Local Thermal Perturbations around a Single CNT

Let us consider the heat transfer problem for a local disc-shaped thermal perturbation of a polymer matrix around a single CNT. This task is associated with the heat transfer problem arising from the isothermal crystallization of the polymer matrix on the surface of CNT in the polymer/CNT nanocomposite. Indeed, the local temperature in the region of crystal birth can be significantly increased due to the heat released at crystallization even under isothermal boundary conditions. In this paper, we focus on the analytical solution of the problem with *dynamic* heat capacity. The aim of this work is to study the *nonequilibrium* thermal response of the polymer matrix at fast local thermal perturbations around CNT in the polymer/CNT nanocomposite. Thus, the difference between the thermal parameters of the crystal and the polymer matrix is neglected. The boundary value problem accounting for this difference will be considered in a separate paper. In addition, the thermal parameters of the polymer matrix are considered independent from the temperature at small thermal perturbations.

The temperature distribution around a single nanotube T(t,r,z) can be described by a nonhomogeneous second-order linear partial differential parabolic equation with two spatial variables; see Equation (3). The analytical solution presented is this paper can be applied to any glass-forming matrix. As an example, for model calculations, thermal parameters close to the parameters of organic glass-forming polymers [48], which are often used as a matrix for nanocomposites, are considered. The thermal parameters used for model calculations are presented in Table 1.

Suppose that the polymer matrix is heated by a heating pulse of duration $\tau_{p}$. Let the heat flux Φ(t,r,z) be distributed uniformly in the disc-shaped region around CNT. This heat flux can be released at crystallization of a disc-shaped polymer crystal nucleated on the CNT surface. Assume that the radius and the thickness of the heating zone are $R_{C}$ and $2L_{C}$, respectively and that the radius of the nanotube equals $R_{1}$; see Figure 1. Thus, Φ(t,r,z) is distributed in the domain $-L_{C}\leq z\leq L_{C}$ and $R_{1}\leq r\leq R_{C}$; see Figure 1. Suppose $\Phi \left(t,r,z\right)=F\left(t\right)\Phi_{0}$, where $\Phi_{0}=\rho h_{0}/\tau_{p}$ with $h_{0}=$ 200 J/g (see Table 1) and F(t) is a unit pulse function: F(t)=1 if $0<t\leq \tau_{p}$ and F(t)=0 otherwise. The temperature of the polymer matrix equals the thermostat temperature $T_{t}$ at a sufficiently large distance from the heating zone. Thus, the heat transfer problem can be calculated in a sufficiently large cylinder with isothermal boundaries. In fact, the response T(t,r,z) practically does not change at a distance of about 100 nm from the center of the heating zone, at least on a nanosecond timescale; see Figures 3,6–8. Therefore, the boundary value problem is considered in cylindrical domains with $R_{1}=$ 5 nm, $R_{2}=$ 150 nm, and $L_{z}=$ 100 nm, as well as $R_{1}=$ 10 nm, $R_{2}=$ 300 nm, and $L_{z}=$ 100 nm. However, the results are verified for domains of different sizes; see Figure 2a. Assume the temperature distribution T(t,r,z) is measured from the temperature of the thermostat $T_{t}$. Thus, $T\left(t,r,\pm L_{z}\right)=0$ and $T\left(t,R_{2},z\right)=0$; see Figure 1. The geometric parameters of the boundary value problem are collected in Table 2. The analytical solution presented in this paper can be applied to the boundary value problem with cylindrical symmetry under various reasonable geometric parameters. In fact, the dynamics of the thermal response T(t,r,z) does not change qualitatively when the geometric parameters change. Further the calculations are performed for $R_{1}$ and $R_{C}$, varying in the range 5–10 nm and 20–50 nm, respectively. Such parameters can be interesting for the analysis of the heat transfer process at the shish-kebab crystal structure formation in CNT/polymer composites. In addition, we focus on the dependence of the fast thermal response T(t,r,z) on the thermal contact conductance and $\lambda_{CNT}$.

The thermal conductivity $\lambda_{CNT}$ of an individual single-walled carbon nanotube (SWCNT) along its axis can be about 3500 $W\cdot m^{-1}K^{-1}$ at room temperature [56,57]. $\lambda_{CNT}$ is determined under the assumption that the wall thickness of the nanotube $b_{CNT}$ is equal to the thickness of a single-layer graphene 0.34 nm [56,57,58,59]. This means that the heat is conducted along the axis of CNT through the area of $\pi d_{CNT}b_{CNT}$, where $d_{CNT}$ is the diameter of CNT. The thermal conductivity of CNT with defects and multi-walled nanotubes (MWCNT) can be lower than 1000 $W\cdot m^{-1}K^{-1}$ [57,58,59]. Moreover, the thermal conductivity of CNT can be significantly reduced by the interaction of CNT with the polymer matrix, similar to that observed in graphene attached to a substrate [57,58]. Next, for model calculations, the thermal conductivity $\lambda_{CNT}$ is considered in the range 100–1000 $W\cdot m^{-1}K^{-1}$ regardless of whether single-walled or multi-walled CNT is dispersed in the polymer matrix. The thermal contact conductance $G_{C}$ between the polymer matrix and the solid surface can be in the range 10^6^–10^8^
$W\cdot m^{-2}K^{-1}$ [60].

Initially, we consider the case of a very perfect thermal contact as well as a very large thermal conductivity $\lambda_{CNT}$. In this case, the temperature on the surface of the nanotube is very close to $T_{t}$, if $\lambda_{CNT}$ is large enough. In fact, $\lambda_{CNT}$ should be at least much larger than $\lambda L_{C}/b_{CNT}=$ 10 $W\cdot m^{-1}K^{-1}$ for $L_{C}=$ 10 nm.

## 4. Dynamics of Temperature Distribution for a Very Large $\lambda_{CNT}$ and Perfect Thermal Contact

Consider the dynamics of the temperature distribution T(t,r,z) in the case of a very large thermal conductivity $\lambda_{CNT}$ when the temperature of the nanotube $T_{CNT}\left(t,z\right)$ is very close to the temperature of the thermostat. Assume an ideal thermal contact of the polymer/CNT interface. Then $T\left(t,R_{1},z\right)=T_{CNT}\left(t,z\right)$. Thus, the boundary value problem can be analyzed over the domain $0\leq z\leq L_{z}$ and $R_{1}\leq r\leq R_{2}$ with the following homogeneous mixed boundary conditions:(5)$$T\left(t,R_{1},z\right)=0,\;T\left(t,R_{2},z\right)=0,\;\mathrm{and}\;T\left(t,r,L_{z}\right)=0$$
(6)∂T(t,r,z)/∂z=0 on the plane z=0

Note that the temperature is counted from the temperature of the thermostat $T_{t}$ and that the zero initial condition (T(t,r,z)=0 if t≤0) is considered. The boundary value problem, associated with Equations (3), (5), and (6), can be solved by separation of variables [61]. Consider the orthogonal functions $\phi_{0}\left(\mu_{m}r/R_{1}\right)=\left(J_{0}\left(\mu_{m}\right)Y_{0}\left(\mu_{m}r/R_{1}\right)-Y_{0}\left(\mu_{m}\right)J_{0}\left(\mu_{m}r/R_{1}\right)\right)$, where $\left\{\mu_{m}\right\}$ is the monotonously increasing sequence of positive (dimensionless) roots of the equation $\phi_{0}\left(\mu_{m}s\right)=0$ at m=1,2,3… and $s=R_{2}/R_{1}$ and where $J_{0}\left(\mu_{m}r/R_{1}\right)$ and $Y_{0}\left(\mu_{m}r/R_{1}\right)$ are zero-order Bessel functions of the first and the second kind, respectively. Note that $\phi_{0}\left(\mu_{m}\right)\equiv 0$. Thus, the solution of the boundary value problem can be presented as a series expansion:(7)$$T\left(t,r,z\right)=\underset{n=0}{\sum }\underset{m=1}{\sum }\psi_{m,n}\left(t\right)\phi_{0}\left(\mu_{m}r/R_{1}\right)cos(\eta_{n}z)$$ where the orthogonal eigenfunction $\phi_{0}\left(\mu_{m}r/R_{1}\right)cos(\eta_{n}z)$ satisfies the boundary conditions at the corresponding eigenvalues $\mu_{m}$ and $\eta_{n}=\pi \left(2n+1\right)/2L_{z}$ for n=0,1,2,…

First, we find the equilibrium thermal response $\tilde{T}\left(t,r,z\right)$ corresponding to the equilibrium heat capacity at $\varepsilon_{0}=0$; see Equation (3). Then, the Fourier components of Equation (3) are equal to
(8)$$\partial \psi_{m,n}\left(t\right)/\partial t+\left({\tilde{\tau }}_{m}^{-1}+\tau_{n}^{-1}\right)\psi_{m,n}\left(t\right)=B_{m,n}\left(t\right)$$
where ${\tilde{\tau }}_{m}^{-1}={\left(\mu_{m}/R_{1}\right)}^{2}D_{0}$, $\tau_{n}^{-1}=\eta_{n}^{2}D_{0}$ and
(9)$$B_{m,n}\left(t\right)=F\left(t\right)\frac{2\Phi_{0}}{L_{z}\rho c_{0}}\int_{0}^{L_{z}}\left(C_{m}\int_{R_{1}}^{R_{2}}\phi_{0}\left(\mu_{m}r/R_{1}\right)rdr\right)cos(\eta_{n}z)dz$$

The normalization factor $C_{m}$ in Equation (9) equals
(10)$$C_{m}=\frac{2R_{1}^{-2}}{{\left(s\phi_{1}\left(\mu_{m}s\right)\right)}^{2}-{\left(\phi_{1}\left(\mu_{m}\right)\right)}^{2}}$$
where $\phi_{1}\left(\mu_{m}r/R_{1}\right)=\left(Y_{0}\left(\mu_{m}\right)J_{1}\left(\mu_{m}r/R_{1}\right)-J_{0}\left(\mu_{m}\right)Y_{1}\left(\mu_{m}r/R_{1}\right)\right)$. After the integration of Equation (9), we get $B_{m,n}\left(t\right)=F\left(t\right)A_{m,n}\Phi_{0}/\rho c_{0}$, where
(11)$$A_{m,n}=\frac{2sin(\eta_{n}L_{C})}{\eta_{n}L_{z}}\cdot \frac{-2}{\mu_{m}}\frac{s_{C}\phi_{1}\left(\mu_{m}s_{C}\right)-\phi_{1}\left(\mu_{m}\right)}{\left[{\left(s\phi_{1}\left(\mu_{m}s\right)\right)}^{2}-{\left(\phi_{1}\left(\mu_{m}\right)\right)}^{2}\right]}$$
where $s_{C}=R_{C}/R_{1}$. The exact solution of Equation (8) equals
(12)$$\psi_{m,n}\left(t\right)=\int_{0}^{t}B_{m,n}\left(t^{\prime }\right)exp\left(-\left({\tilde{\tau }}_{m}^{-1}+\tau_{n}^{-1}\right)\left(t-t^{\prime }\right)\right)dt^{\prime }$$

Therefore,
(13)$$\tilde{T}\left(t,r,z\right)=\underset{n=0}{\sum }\underset{m=1}{\sum }\phi_{0}\left(\mu_{m}r/R_{1}\right)cos(\eta_{n}z)\int_{0}^{t}B_{m,n}\left(t^{\prime }\right)exp\left(-\left({\tilde{\tau }}_{m}^{-1}+\tau_{n}^{-1}\right)\left(t-t^{\prime }\right)\right)dt^{\prime }$$

After integrating Equation (13) for the pulse function $F\left(t\right)=\theta \left(t\right)\left(1-\theta \left(t-\tau_{p}\right)\right)$, where θ(t) is the Heaviside unit step function at zero convention θ(t)=0, we find
(14)$$\tilde{T}\left(t,r,z\right)=\underset{n=0}{\sum }\underset{m=1}{\sum }{\tilde{\Gamma }}_{m,n}\left(t\right)\phi_{0}\left(\mu_{m}r/R_{1}\right)cos(\eta_{n}z)A_{m,n}\Phi_{0}/\rho c_{0}$$
where ${\tilde{\Gamma }}_{m,n}\left(t\right)=\left[\left(1-exp(-t\left({\tilde{\tau }}_{m}^{-1}+\tau_{n}^{-1}\right))\right)-\left(1-exp(-\left(t-\tau_{p}\right)\left({\tilde{\tau }}_{m}^{-1}+\tau_{n}^{-1}\right))\right)\theta \left(t-\tau_{p}\right)\right]/\left({\tilde{\tau }}_{m}^{-1}+\tau_{n}^{-1}\right)$.

The solution of the boundary value problem with dynamic heat capacity for positive $\varepsilon_{0}$ and $\tau_{0}$ can be found similarly; for details see Appendix A. Next, as an example, the calculations are performed for $\varepsilon_{0}=$ 1/3 and different $\tau_{0}$. The boundary value problem is considered in cylindrical domains at $R_{1}=$ 5 nm, $R_{2}=$ 150 nm, and $L_{z}=$ 100 nm as well as at 10 nm, $R_{2}=$ 300 nm, and $L_{z}=$ 100 nm. Note that the thermal response of the polymer matrix T(t,r,z) is counted further from the temperature of the thermostat $T_{t}$. The analytical solution is presented as a series expansion. The temperature distribution T(t,r,z) can be accurately calculated if we take into account the sufficiently large number N of the first members of the series. In fact, the calculation accuracy within 0.2% and 0.05% error is achieved at N=50 and N=100, respectively. Calculations at N=200 do not change the results within 0.05% error. Further calculations are performed at N=100.

Let us consider the equilibrium $\tilde{T}\left(t,r,z\right)$ and nonequilibrium T(t,r,z) thermal response for $\tau_{0}=$ 1 ns, 3 ns, 10 ns, and 30 ns. As an example, suppose that $\tau_{p}=$ 2 ns, $R_{1}=$ 5 nm, $L_{C}=$ 10 nm, and $R_{C}=$ 50 nm or 20 nm. The calculations are performed in the domain with isothermal walls at 150 nm and $L_{z}=$ 100 nm. Note that the result is the same for a twice larger domain with $R_{2}=$ 300 nm and $L_{z}=$ 200 nm; see Figure 2a. Indeed, the response T(t,r,z) practically does not change at a distance of about 100 nm from the center of the heating region; see Figure 3. Thus, the result is independent from the position of the boundaries if the boundaries are located at a sufficiently large distance from the center of the heating zone. However, T(t,r,z) depends on the geometric parameters $R_{C}$, $L_{C}$, and $R_{1}$; see Figure 2 and Figure 4. As an example, we consider the temperature distributions in the middle of the heating zone T(t,r,0) and $T\left(t,R_{C}/2,z\right)$. As expected, the time dependence $T\left(t,R_{C}/2,0\right)$ is saturated at t of the order of $\tau_{C}=R_{C}^{2}/4D_{0}$; see Figure 2. In fact, $\tau_{C}$ is about 1 ns and 4 ns for $R_{C}=$ 20 nm and 50 nm, respectively.

The thermal response of the polymer matrix with delayed dynamic heat capacity is larger than the equilibrium response in the early stages of the heating process; see Figure 2 and Figure 4. It is notable that even fast components of the dynamic heat capacity (with $\tau_{0}$ about 1 ns) are significant. Nonequilibrium thermal response T(t,r,z) increases with increasing $\tau_{0}$. However, this effect is saturated with the growth of $\tau_{0}$; see Figure 3 and Figure 4. This saturation is observed at lower $\tau_{0}$ in regions of smaller radius $R_{C}$ because smaller regions relax faster to equilibrium with the characteristic relaxation time $\tau_{C}=R_{C}^{2}/4D_{0}$; see Figure 2 and Figure 4.

The effect of dynamic heat capacity is pronounced at early stages of the heating process. Denote by δT(t,r,z) the difference between equilibrium and nonequilibrium response $T\left(t,r,z\right)-\tilde{T}\left(t,r,z\right)$. Consider the relative effect of the dynamic heat capacity on the thermal response. This effect can be described by the ratio $\delta T\left(t,r,z\right)/\tilde{T}\left(t,r,z\right)$. The relative contribution of the nonequilibrium response tends to a constant level at t→0; see Figure 4. As expected, this level increases with $\varepsilon_{0}$.

Thus, the dynamic heat capacity significantly affects the thermal response of the polymer matrix to local fast thermal perturbations, especially at the initial stages of the heating process. This effect depends on $\tau_{0}$ and $\varepsilon_{0}$, as well as the size of the heating zone.

The spectrum of relaxation times $\tau_{0}$ of the dynamic heat capacity $c_{dyn}\left(t\right)$ of the polymer matrix strongly depends on the temperature, especially near the glass transition temperature. Denote by $\tau_{AV}\left(T\right)$ the average relaxation time $\tau_{AV}\left(T\right)=\int_{0}^{\infty }\tau_{0}H\left(\tau_{0},T\right)d\tau_{0}$. In fact, $\tau_{AV}\left(T\right)$ is about $1/\omega_{max}$, where $\omega_{max}$ is the angular frequency corresponding to the maximum of the imaginary part of the dynamic heat capacity; for details, see Reference [14]. Denote by ${\tilde{\tau }}_{AV}\left(T\right)=1/\omega_{max}$. Then, ${\tilde{\tau }}_{AV}\left(T\right)$ can be obtained from the empirical Vogel–Fulcher–Tammann–Hesse (VFTH) relationship:(15)$$\log(\omega_{max})=A-B/\left(T-T_{0}\right)$$

The parameters of Equation (15) can be specified using the results of broadband dielectric and heat capacity spectroscopy. As an example, we get for polystyrene the following: *A* = 10.2, *B* = 388 K, and *T*_0_ = 341.5 K—obtained from heat capacity spectroscopy—and *A* = 10.5, *B* = 475.3 K, and *T*_0_ = 334.4 K—from dielectric spectroscopy. We also get for PMMA the following: *A* = 7.3, *B* = 185 K, and *T*_0_ = 354.3 K—from dielectric spectroscopy [18].

It is noteworthy that the average relaxation time ${\tilde{\tau }}_{AV}\left(T\right)$ for polymers exceeds 10 ns in a wide temperatures range above the glass transition temperature; see Figure 5. However, the effect of the temporal dispersion of the dynamic heat capacity is saturated above 10 ns for nanometer scale regions; see Figure 3 and Figure 4. Therefore, the effect of dynamic heat capacity on the fast thermal response of the polymer matrix can be estimated for $\tau_{0}=$ 10 ns if $R_{C}$ is about several tens of nm. Indeed, the effect is almost the same for larger $\tau_{0}$; see Figure 3 and Figure 4. In fact, the shape of the distribution function $H\left(\tau_{0},T\right)$ does not significantly affect the thermal response T(t,r,z) [13,14].

Summarizing, it can be concluded that the local overheating can be significantly enhanced even at high temperatures due to the very fast components (with $\tau_{0}$ about 10 ns) of the dynamic heat capacity; see Figure 3, Figure 4, and Figure 5. Next, the temperature distribution around CNT with limited $G_{C}$ and $\lambda_{CNT}$ is studied.

## 5. Dynamics of Temperature Distribution around CNT at Different $G_{C}$ and $\lambda_{CNT}$

Consider the dynamics of the temperature distribution T(t,r,z) in the case of limited thermal contact conductance $G_{C}$ and thermal conductivity $\lambda_{CNT}$. The temperature on the polymer/CNT interface has a step due to the thermal contact resistance $G_{C}^{-1}$ of the polymer/CNT interface:(16)$$T\left(t,R_{1},z\right)-T_{CNT}\left(t,z\right)=q\left(t,z\right)/G_{C}$$ where the heat flux between the polymer matrix and CNT is $q\left(t,z\right)=\lambda {\left(\partial T\left(t,r,z\right)/\partial r\right)}_{|R_{1}}$. The energy balance equation at the polymer/CNT interface is
(17)$$\lambda {\left(\partial T\left(t,r,z\right)/\partial r\right)}_{|R_{1}}+\lambda_{CNT}b_{CNT}\partial^{2}T_{CNT}\left(t,z\right)/\partial z^{2}=0$$
where $T_{CNT}\left(t,z\right)$ can be presented as a series expansion $T_{CNT}\left(t,z\right)=\underset{n=0}{\sum }\chi_{n}\left(t\right)cos(\eta_{n}z)$ consistent with the boundary conditions of Equations (5) and (6). The boundary condition at the polymer/CNT interface can be presented in the form $\lambda {\left(\partial T\left(t,r,z\right)/\partial r\right)}_{|R_{1}}=\frac{1}{G_{C}^{-1}+G_{CNT}^{-1}}T\left(t,R_{1},z\right)$, where the thermal conductance $G_{CNT}$ of CNT along z-axis is of the order of $\lambda_{CNT}b_{CNT}L_{C}^{-2}$. Indeed, the main contribution to the gradient $\partial T_{CNT}\left(t,z\right)/\partial z$ is of the order of $T_{CNT}\left(t,z\right)/L_{C}$. Thus, $\partial^{2}T_{CNT}\left(t,z\right)/\partial z^{2}$ can be approximated by $T_{CNT}\left(t,z\right)/L_{C}^{2}$ or $\eta_{2}^{2}T_{CNT}\left(t,z\right)$ for $\eta_{2}L_{C}\approx 1$. Thus, we get $G_{CNT}=\lambda_{CNT}b_{CNT}\eta_{2}^{2}$. Note that $G_{CNT}$ is about 3∙10^9^
$W\cdot m^{-2}K^{-1}$ at $\lambda_{CNT}=$ 10^3^
$W\cdot m^{-1}K^{-1}$ and $L_{C}=$ 10 nm. Therefore, $G_{C}^{-1}>>G_{CNT}^{-1}$, since the thermal contact conductance $G_{C}$ for polymer/solid interface can be in the range 10^6^–10^8^
$W\cdot m^{-2}K^{-1}$ [60]. Consequently, the error of the $G_{CNT}$ estimate has an insignificant effect on the factor $\frac{1}{G_{C}^{-1}+G_{CNT}^{-1}}$, which varies within 2.5% at $G_{C}=$ 10^8^
$W\cdot m^{-2}K^{-1}$ if factor $\eta_{2}^{2}$ in $G_{CNT}=\lambda_{CNT}b_{CNT}\eta_{2}^{2}$ is replaced by, say, factor $\eta_{3}^{2}$. Furthermore, $T_{CNT}\left(t,z\right)$ is much lower than the temperature of the polymer matrix in the middle of the heating zone (see Figure 6) and even $T_{CNT}\left(t,z\right)$ << $T\left(t,R_{1},z\right)$ at $G_{C}\leq$ 10^8^
$W\cdot m^{-2}K^{-1}$ (see Figure 7). Thus, the error in the $T_{CNT}\left(t,z\right)$ approximation insignificantly affects the temperature distribution T(t,r,z). Then, the energy balance of Equation (17) can be presented as $\lambda {\left(\partial T\left(t,r,z\right)/\partial r\right)}_{|R_{1}}=G_{CNT}T_{CNT}\left(t,z\right)$ with $G_{CNT}=\lambda_{CNT}b_{CNT}\eta_{2}^{2}$. Additionally, we get the following from Equation (16): $T\left(t,R_{1},z\right)-T_{CNT}\left(t,z\right)=T_{CNT}\left(t,z\right)G_{CNT}/G_{C}$. Therefore, the boundary condition at the polymer/CNT interface is
(18)$${\left(\partial T\left(t,r,z\right)/\partial r\right)}_{|R_{1}}=k_{C}T\left(t,R_{1},z\right)$$
where $k_{C}=\frac{\lambda^{-1}}{G_{C}^{-1}+G_{CNT}^{-1}}$. Thus, the boundary value problem can be analyzed over the domain $0\leq z\leq L_{z}$ and $R_{1}\leq r\leq R_{2}$ with the following mixed boundary conditions:(19)$$T\left(t,R_{2},z\right)=0\;\mathrm{and}\;T\left(t,r,L_{z}\right)=0$$
(20)$${\left(\partial T\left(t,r,z\right)/\partial r\right)}_{|R_{1}}=k_{C}T\left(t,R_{1},z\right)\;\mathrm{and}\;{\left(\partial T\left(t,r,z\right)/\partial z\right)}_{|Z=0}=0$$

The boundary value problem of Equations (3), (19), and (20) can be solved similar to the problem considered in Section 4; for details, see Appendix B.

First, we compare the results obtained in the previous section for the ideal case of extremely large $G_{C}$ and $\lambda_{CNT}$ when $T_{CNT}$ and $T\left(t,R_{1},z\right)$ are equal to the thermostat temperature with the temperature distribution T(t,r,z) for large but limited 10^9^
$W\cdot m^{-2}K^{-1}$ and $\lambda_{CNT}=$ 10^3^
$W\cdot m^{-2}K^{-1}$. As expected, the temperature distributions are practically the same in both cases; see Figure 6. However, these solutions are obtained from quite different boundary value problems. As expected, the temperature distributions $T\left(t,R_{1},z\right)$ and $T_{CNT}\left(t,z\right)$ are much lower than the temperature of the polymer matrix in the middle of the heating zone; see Figure 6.

Next, consider the effect of the thermal contact conductance $G_{C}$ on the temperature distribution in the polymer matrix around CNT at $\lambda_{CNT}=$ 10^3^
$W\cdot m^{-1}K^{-1}$. Note that the temperature distribution $T\left(t,R_{1},z\right)$ tends to $T\left(t,R_{C}/2,z\right)$ with a decrease in the thermal contact conductance $G_{C}$; see Figure 7 and Figure 8. In fact, the difference $T\left(t,R_{C}/2,z\right)-T\left(t,R_{1}/2,z\right)$ is insignificant at $G_{C}\leq$ 10^6^
$W\cdot m^{-2}K^{-1}$. Thus, the thermal contact with $G_{C}\leq$ 10^6^
$W\cdot m^{-2}K^{-1}$ can be considered an almost thermally isolating contact for fast thermal perturbations.

Next, consider the effect of the thermal conductivity $\lambda_{CNT}$ on the temperature distribution in the polymer matrix around CNT at different $G_{C}$; see Figure 8. The effect of CNT with $\lambda_{CNT}=$ 10^2^
$W\cdot m^{-1}K^{-1}$ on the dynamics of the temperature distribution is as strong as with $\lambda_{CNT}=$ 10^3^
$W\cdot m^{-1}K^{-1}$. In fact, $\lambda_{CNT}$ in the range 10^2^–10^3^
$W\cdot m^{-1}K^{-1}$ is large enough to significantly affect the nanoscale heat conduction of the polymer/CNT composites.

The heat flux removed from the heating zone by CNT decreases with a decrease of the thermal contact conductance $G_{C}$. Denote by $P_{CNT}$ the heat flux from the heated zone into CNT, say, at $t=\tau_{p}/2$ and $\tau_{p}=$ 2 ns. This heat flux can be estimated as $P_{CNT}=4\pi R_{1}L_{C}\lambda {\left(\partial T\left(\tau_{p}/2,r,0\right)/\partial r\right)}_{|R_{1}}$. The heat power released in the volume $V_{C}=2\pi \left(R_{C}^{2}-R_{1}^{2}\right)L_{C}$ is equal to $P_{H}=\Phi_{0}V_{C}$. Consider the ratio $P_{CNT}/P_{H}$. This ratio equals about 11%, 8%, and 2% at $G_{C}=$ 10^9^
$W\cdot m^{-2}K^{-1}$, 10^8^
$W\cdot m^{-2}K^{-1}$, and 10^7^
$W\cdot m^{-2}K^{-1}$, respectively. However, this ratio increases for smaller $R_{C}$; see Figure 9 at $R_{C}=$ 20 nm. Thus, the influence of CNT on the heat transfer in the composite at small sizes of the heating zone at $R_{C}=$ 20 nm is significant; see Figure 9.

We can now estimate the characteristic length $T{\left(\partial T/\partial x\right)}^{-1}$ of the temperature gradients in the polymer matrix, which is considered to be longer than the phonon mean-free-path in the polymer. The maximum gradient exists near the polymer/CNT interface at $r=R_{1}$ in the middle of the heating zone at z=0 and at the end of the heating pulse at $t=\tau_{p}$. Thus, ${\left(\partial T\left(\tau_{p},r,0\right)/\partial r\right)}_{|R_{1}}=$ 5.6∙10^9^ K/m at $G_{C}=$ 10^9^, $\tau_{p}=$ 2 ns, $R_{C}=$ 50 nm, $L_{C}=$ 10 nm, $R_{1}=$ 10 nm, $R_{2}=$ 300 nm, and $L_{z}=$ 100 nm. Then the length $T/{\left(\partial T\left(\tau_{p},r,0\right)/\partial r\right)}_{|R_{1}}$ is about 100 nm at T about 400 K. This length is even more than 300 nm at $G_{C}=$ 10^7^
$W\cdot m^{-2}K^{-1}$. Thus, the phonon mean-free-path in the polymer matrix is much less than the characteristic length of the temperature gradients considered in this paper.

Summarizing, it can be concluded that the heat conduction in the polymer/CNT composites significantly depends on the thermal contact conductance at $G_{C}$ in the range 10^7^–10^8^
$W\cdot m^{-2}K^{-1}$; see Figure 7. However, CNT has little effect on the temperature distribution in the polymer matrix at $G_{C}<$ 10^7^
$W\cdot m^{-2}K^{-1}$; see Figure 7. Thermal contact with $G_{C}$ about 10^9^
$W\cdot m^{-2}K^{-1}$ can be considered ideal contact; see Figure 6. The thermal conductivity $\lambda_{CNT}$ in the range 10^2^–10^3^
$W\cdot m^{-1}K^{-1}$ is large enough to significantly affect the dynamics of the heat conduction in the polymer/CNT composites; see Figure 8 and Figure 9. The relative effect of CNT on the heat conduction is more pronounced for the heating zone of small sizes; see Figure 9 at $R_{C}=$ 20 nm.

## 6. Conclusions

The classical theory of heat transfer is insufficient to describe the fast heat conduction processes in polymer/CNT nanocomposites. Relaxation processes associated with the dynamic heat capacity $c_{dyn}\left(t\right)$ are very important at fast thermal perturbations in the nanocomposites. Nonequilibrium dynamics of polymer/CNT nanocomposites in nanosecond and longer timescales can be described by linear integrodifferential equations. The thermal response T(t,r,z) of the polymer matrix in polymer/CNT nanocomposites can be calculated analytically for local thermal perturbations around CNT at cylindrical geometry. Thus, an analytical solution for the nonequilibrium thermal response of the polymer matrix is obtained for different parameters of CNT and thermal-contact conductance $G_{C}$ of the polymer/CNT interface.

In fact, the dynamic heat capacity $c_{dyn}\left(t\right)$ of the polymer matrix lags behind the heat capacity of an ideal equilibrium material. Therefore, the thermal response T(t,r,z) is higher than that of the equilibrium substance, mainly at the early stages of the heating process. It is remarkable that even fast components of $c_{dyn}\left(t\right)$ (with relaxation time about 1 ns) significantly affect the thermal response to *local* thermal perturbations at the nanometer scale. However, the effect of the temporal dispersion of the dynamic heat capacity $c_{dyn}\left(t\right)$ on the thermal response T(t,r,z) is saturated at $\tau_{0}$ exceeding several tens of ns if the size of the local heating zone is about several tens of nm.

The spectrum of relaxation times $\tau_{0}$ of the dynamic heat capacity $c_{dyn}\left(t\right)$ of the polymer matrix depends on temperature, especially near the glass transition temperature where the relaxation times become very long. Nevertheless, the average relaxation time in glass-forming polymers, usually used as a polymer matrix in nanocomposites, exceeds 10 ns in a wide temperatures range above the glass transition temperature. Therefore, the effect of the temporal dispersion of the dynamic heat capacity $c_{dyn}\left(t\right)$ on the thermal response T(t,r,z) can be significant even at temperatures considerably higher than the glass transition temperature. Thus, the local overheating of the polymer matrix in the composite can be significantly enhanced even at high temperatures due to the fast components (with $\tau_{0}$ about 10 ns) of the dynamic heat capacity.

The effect of the thermal contact conductance $G_{C}$ on the dynamics of temperature distribution in the polymer matrix around CNT is significant at $G_{C}$ in the range 10^7^–10^8^
$W\cdot m^{-2}K^{-1}$. However, CNT has little effect on the temperature distribution at $G_{C}<$ 10^7^
$W\cdot m^{-2}K^{-1}$. The thermal conductivity of CNT in the range 10^2^–10^3^
$W\cdot m^{-1}K^{-1}$ is large enough to significantly affect the heat conduction in the polymer/CNT composites. The obtained results can be useful for the analysis of the heat transfer process at the early stages of crystallization in CNT/polymer nanocomposites.

## Figures and Tables

**Figure 1 molecules-24-02794-f001:**
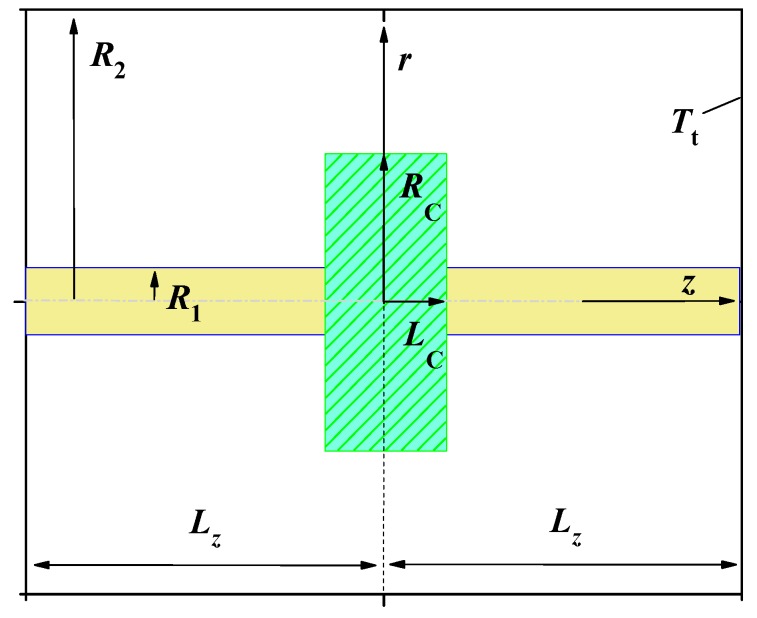
Disc-shaped heating zone around CNT (not to scale).

**Figure 2 molecules-24-02794-f002:**
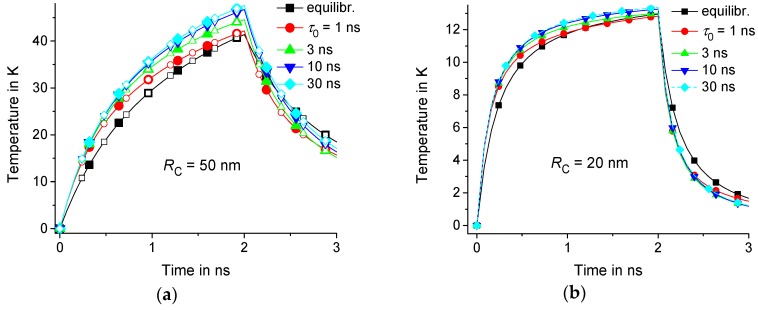
Time dependence of equilibrium solution $\tilde{T}\left(t,R_{C}/2,0\right)$ is represented by the lines marked by squares and those at nonequilibrium $T\left(t,R_{C}/2,0\right)$ at $\tau_{0}=$ 1 ns, 3 ns, 10 ns, and 30 ns are respectively represented by circles, upwards-facing triangles, downwards-facing triangles, and diamonds for $R_{C}=$ 50 nm (**a**) and 20 nm (**b**); $R_{1}=$ 5 nm, $R_{2}=$ 150 nm, and $L_{z}=$ 100 nm are represented by the filled symbols, as well as $R_{2}=$ 300 nm and $L_{z}=$ 200 nm are represented by the open symbols. Note that the temperature is counted from $T_{t}$.

**Figure 3 molecules-24-02794-f003:**
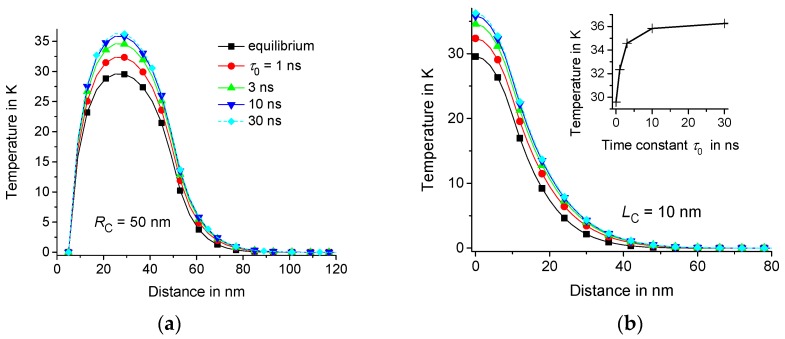
Temperature distribution T(t,r,0) vs. r (**a**) and $T\left(t,R_{C}/2,z\right)$ vs. z (**b**) at t= 1 ns, $\tau_{p}=$ 2 ns, $R_{C}=$ 50 nm, $L_{C}=$ 10 nm, $R_{1}=$ 5 nm, $R_{2}=$ 150 nm, and $L_{z}=$ 100 nm. The equilibrium solution is represented by lines marked by squares and the nonequilibrium solutions at $\tau_{0}=$ 1 ns, 3 ns, 10 ns, and 30 ns are represented by circles, upwards-facing triangles, downwards-facing triangles, and diamonds, respectively. $T\left(t,R_{C}/2,0\right)$ vs. $\tau_{0}$ at t= 1 ns is shown in the insert of Figure 3b.

**Figure 4 molecules-24-02794-f004:**
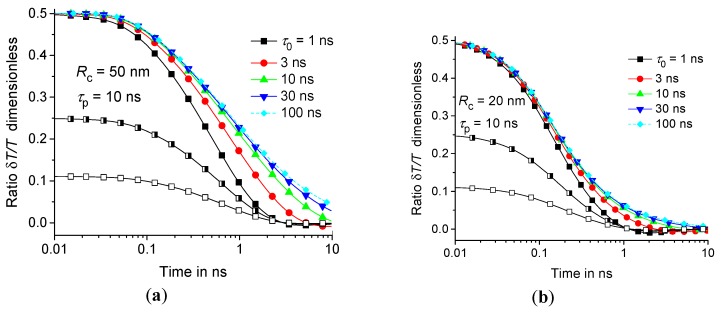
Time dependence of the ratio $\delta T\left(t,R_{C}/2,0\right)/\tilde{T}\left(t,R_{C}/2,0\right)$ for $R_{C}=$ 50 nm (**a**) and 20 nm (**b**) at $\tau_{p}=$ 10 ns and at $\tau_{0}=$ 1 ns, 3 ns, 10 ns, 30 ns, and 100 ns—the squares, circles, upwards-facing triangles, downwards-facing triangles, and diamonds, respectively; $\varepsilon_{0}=$ 1/3, 0.2, and 0.1 are represented by filled, semi-filled, and open symbols, respectively. The geometric parameters are the same as in Figure 3.

**Figure 5 molecules-24-02794-f005:**
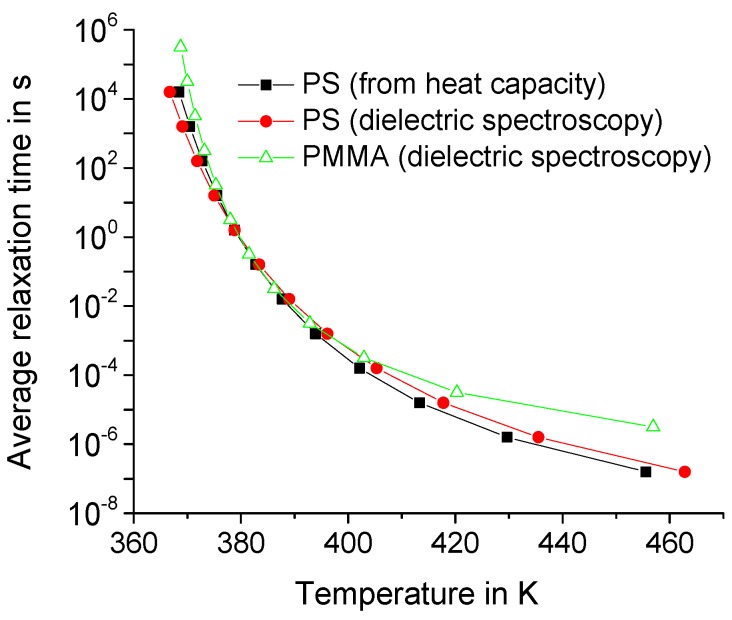
Temperature dependence of the average relaxation time ${\tilde{\tau }}_{AV}\left(T\right)$ for polystyrene (PS) and poly(methyl methacrylate) (PMMA), the filled and open symbols, respectively.

**Figure 6 molecules-24-02794-f006:**
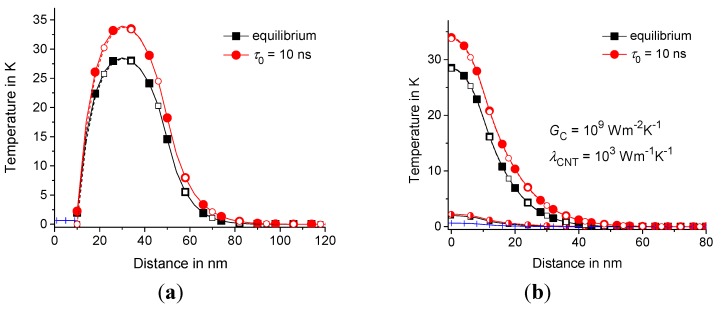
Temperature distribution T(t,r,z) vs. r (**a**) and $T\left(t,R_{C}/2,z\right)$ vs. z (**b**) are represented by the filled symbols, $T_{CNT}\left(t,z\right)$ is represented by the crosses, and $T\left(t,R_{1},z\right)$ is represented by the semi-filled symbols at $G_{C}=$ 10^9^
$W\cdot m^{-2}K^{-1}$ and $\lambda_{CNT}=$ 10^3^
$W\cdot m^{-1}K^{-1}$. The solution obtained in Section 4 for $T\left(t,R_{1},z\right)=0$ is represented by open symbols. The equilibrium and nonequilibrium solutions ($\tau_{0}=$ 10 ns) are represented by squares and circles, respectively; t= 1 ns, $\tau_{p}=$ 2 ns, $R_{C}=$ 50 nm, $L_{C}=$ 10 nm, $R_{1}=$ 10 nm, $R_{2}=$ 300 nm, and $L_{z}=$ 100 nm.

**Figure 7 molecules-24-02794-f007:**
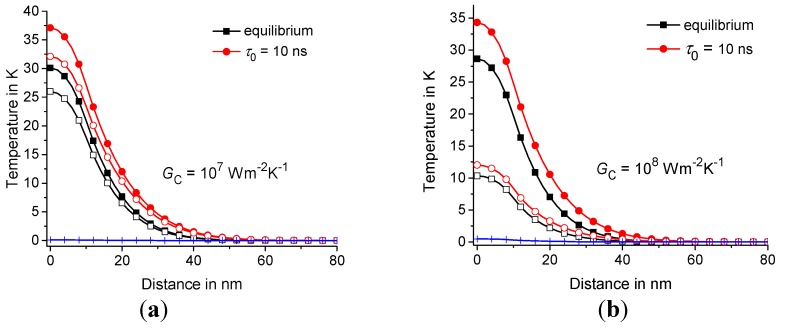
Temperature distribution along the z-axis at $G_{C}=$ 10^7^ W/m^2^K (**a**) and 10^8^ W/m^2^K (**b**). $T\left(t,R_{C}/2,z\right)$ and $T\left(t,R_{1},z\right)$ are represented by the filled and open symbols, respectively, and $T_{CNT}\left(t,z\right)$ is represented by the crosses. The equilibrium and nonequilibrium solutions at $\tau_{0}=$ 10 ns are represented by squares and circles, respectively; t= 1 ns and $\tau_{p}=$ 2 ns. The geometric parameters are the same as in Figure 6.

**Figure 8 molecules-24-02794-f008:**
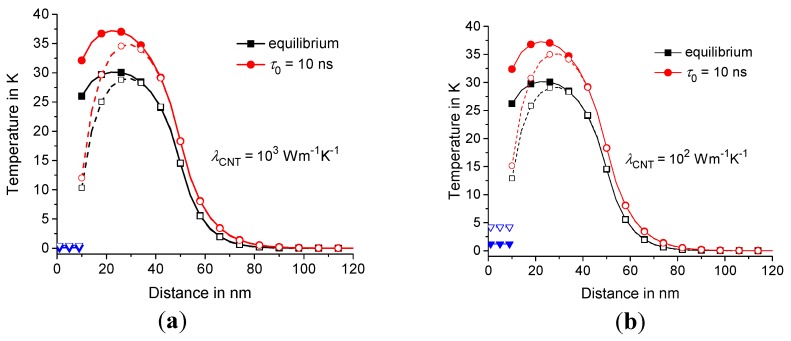
Temperature distribution T(t,r,0) vs. r at $\lambda_{CNT}=$ 10^3^
$W\cdot m^{-1}K^{-1}$ (**a**) and 10^2^
$W\cdot m^{-1}K^{-1}$ (**b**) for $G_{C}=$ 10^7^
$W\cdot m^{-2}K^{-1}$ and 10^8^
$W\cdot m^{-2}K^{-1}$, represented by the filled and open symbols, as well as $T_{CNT}\left(t,z\right)$, represented by the triangles. The equilibrium and nonequilibrium solutions at $\tau_{0}=$ 10 ns are represented by the squares and circles, respectively; t= 1 ns and $\tau_{p}=$ 2 ns. The geometric parameters are the same as in Figure 6.

**Figure 9 molecules-24-02794-f009:**
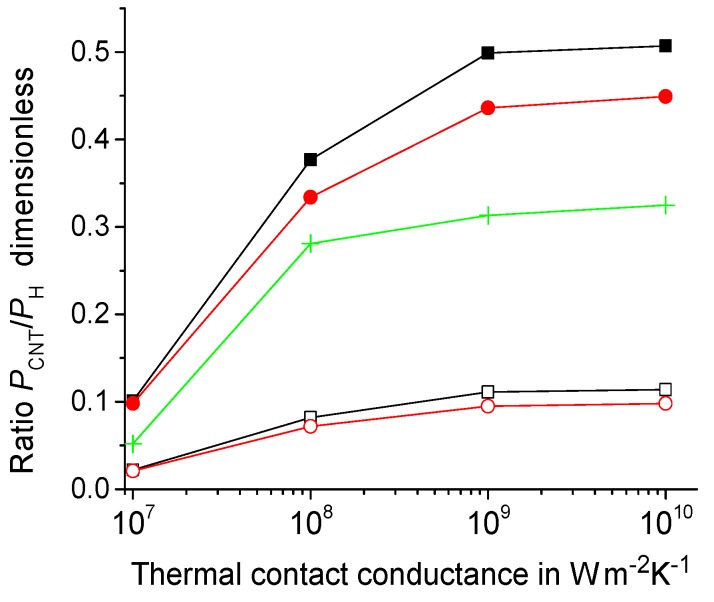
Ratio $P_{CNT}/P_{H}$ vs. $G_{C}$ at $R_{C}=$ 20 nm and 50 nm, represented by the filled and open symbols, respectively, for $\lambda_{CNT}=$ 10^3^
$W\cdot m^{-1}K^{-1}$ and 10^2^
$W\cdot m^{-1}K^{-1}$ (the squares and circles, respectively) at $R_{1}=$ 10 nm as well as at $R_{1}=$ 5 nm for $R_{C}=$ 20 nm and $\lambda_{CNT}=$ 10^3^
$W\cdot m^{-1}K^{-1}$ (the crosses).

**Table 1 molecules-24-02794-t001:** Typical thermal parameters of a polymer matrix (at room temperature and normal pressure).

Density ρ in g/cm^3^	Specific heat capacity $c_{0}$ in J/g·K	Volumetric heat capacity $\rho c_{0}$ in J/m^3^·K	Thermal conductivity λ in $W\cdot m^{-1}K^{-1}$	Thermal diffusivity $D_{0}=\lambda /\rho c_{0}$ in m^2^/s	Heat release at crystallization $h_{0}$ in J/g
1	2	2 × 10^6^	0.3	1.5 × 10^−7^	200

**Table 2 molecules-24-02794-t002:** Geometric parameters of the boundary value problem.

Half thickness of heating zone $L_{C}$ in nm	Radius of heating zone $R_{C}$ in nm	Radius of CNT $R_{1}$ in nm	Distance to thermostat along *r*-axis $R_{2}$ in nm	Distance to thermostat along *z*-axis $L_{z}$ in nm	Ratio $s_{C}=R_{C}/R_{1}$ Dimension-less	Ratio $s=R_{2}/R_{1}$ Dimension-less
10	20–50	5–10	150–300	100	2–10	30

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
