# Peer review of "Nanoscale Heat Conduction in CNT-POLYMER Nanocomposites at Fast Thermal Perturbations"

_molecules, 2019, doi:10.3390/molecules24152794_

Round 1

Reviewer 1 Report

Alexander A. Minakov and Christoph Schick reported a study on "Nanoscale heat conduction in CNT-polymer nanocomposites at fast thermal perturbations", the paper is worth publishing in Molecules, after minor revisions.

The paper is well written, and the topic is very interesting, however the authors show a poor knowledge of polymers, and even if the work is strictly theoretical, they should be more accurate when referring to these materials.

Page 4/5: What does it mean 'typical Polymer'?, In the wide world of polymers nothing is typical, or maybe the authors refer to some 'selected polymers' of Ref.49? The authors should explain better and indicate the pages of reference 49 where they found the data reported in Table 1.

The data reported in Table 1 are used in their equations and all their conclusions could change in changing them, so this referee suggests to indicate clearly to which polymers (o polymers family) their refer.

In conclusion, this referee believes that the subject of this paper is appropriate for Molecules and that it is of some interest, but the paper should be published only after minor, but nevertheless, fundamental revisions.

Reviewer 2 Report

The manuscript of Minakov and Schick deals with non-equilibrium heat transport across a polymer-CNT interfase. Although the topic of non-equilibrium thermal transport is well identified and of current general interest. The specific studied case has no general physical validity. In fact, it becomes almost impossible to related the current results with any sort of real experiment. In addition, most approximations are not sufficiently justified. Furthermore, the authors claim that an analytic solution to the problem is presented which, at least to me, is lost within the lines in the current version of the manuscript. A couple of specific comments: 1) Why do the authors decided to use the parabolic form of the heat equation and not the Maxwell-Cattaneo approach?. This should be at least briefly discussed in the manuscript. 2) Eq 2.1 is introduced as the solution to the problem and is referenced to their own previous work. This is could be confusing to the non-expert reader, i.e., this equation is obviously based on the parabolic heat equation. Why this is not presented as such plus introducing the time dependent heat capacity term?. 3) The following assumptions are used ? > 1 ns ,and on length scales of more than 10 nm and 1 nm for the semi-crystalline and amorphous matrices. All this approximations are not well explained given their large importance within the presented work. 4) The chosen geometry in Fig. 1 seems rather arbitrary. Why the authors choose this precise configuration?. Actually, the reader could profit much more from comparing the output of several geometries which could occur in nature, thus, doable in experiments. 5) The manuscript present a large number of equations which are, in most cases, quite basic. However, for the non expert reader this provides no aggregate value to the fundamental understanding. Most of the simple maths here presented could just be included into supporting information since there is almost no novelty in these equations. However, this is only an opinion and it is fully on hands of the authors of course. 6) No details on the calculations are given!. This is quite surprising since most of the figures presented are the output of numerically solving some of the presented equations. Thus, how did the authors arrived to these solutions, finite element modelling?. In fact, it is confusing that the authors mention that the "analytic solution" is given by Eq. (2.4), however, I see an infinite integral within this equation. In the case that no finite element simulations were conducted, well, the presented geometry could be easily addressed using, e.g., COMSOL multiphysics or similar.
